# Biodegradation and Metabolic Pathway of the Neonicotinoid Insecticide Thiamethoxam by *Labrys portucalensis* F11

**DOI:** 10.3390/ijms232214326

**Published:** 2022-11-18

**Authors:** Oumeima Boufercha, Ana R. Monforte, Allaoueddine Boudemagh, António C. Ferreira, Paula M. L. Castro, Irina S. Moreira

**Affiliations:** 1Laboratory of Molecular and Cellular Biology, University of Brothers Mentouri, Constantine 1, Chaâbat Erssas Campus, Ain El Bey Road, Constantine 25000, Algeria; 2Department of Microbiology, Faculty of Natural and Life Sciences, University of Brothers Mentouri, Constantine 1, Ain El Bey Road, Constantine 25000, Algeria; 3CBQF—Centro de Biotecnologia e Química Fina, Laboratório Associado, Escola Superior de Biotecnologia, Universidade Católica Portuguesa, Rua Diogo Botelho 1327, 4169-005 Porto, Portugal

**Keywords:** thiamethoxam, *Labrys portucalensis* F11, biodegradation, metabolites, toxicity

## Abstract

Thiamethoxam (TMX) is an effective neonicotinoid insecticide. However, its widespread use is detrimental to non-targeted organisms and water systems. This study investigates the biodegradation of this insecticide by *Labrys portucalensis* F11. After 30 days of incubation in mineral salt medium, *L. portucalensis* F11 was able to remove 41%, 35% and 100% of a supplied amount of TMX (10.8 mg L^−1^) provided as the sole carbon and nitrogen source, the sole carbon and sulfur source and as the sole carbon source, respectively. Periodic feeding with sodium acetate as the supplementary carbon source resulted in faster degradation of TMX (10.8 mg L^−1^); more than 90% was removed in 3 days. The detection and identification of biodegradation intermediates was performed by UPLC-QTOF/MS/MS. The chemical structure of 12 metabolites is proposed. Nitro reduction, oxadiazine ring cleavage and dechlorination are the main degradation pathways proposed. After biodegradation, toxicity was removed as indicated using *Aliivibrio fischeri* and by assessing the synthesis of an inducible β-galactosidase by an *E. coli* mutant (Toxi-Chromo test). *L. portucalensis* F11 was able to degrade TMX under different conditions and could be effective in bioremediation strategies.

## 1. Introduction

Insecticides are a widespread class of pesticides designed to kill insects, accounting for 80% of overall pesticide production [1]. Various classes of insecticides such as carbamates, organophosphates, organochlorines and pyrethroids are considered as the main cause of serious or even fatal poisoning in humans, which has led to the creation of a new class of less toxic insecticides called neonicotinoids [2].

Neonicotinoids are chemical insecticides derived from the modification of nicotine [3]. Neonicotinoids appeared in the 1990s [4], and due to their promising insecticidal effect and broad-spectrum against chewing and sucking pests [5], their use has been recorded in over 120 countries and in more than 140 crops [6]. The global insecticide market of neonicotinoids, including imidacloprid, acetamiprid, nitenpyram, thiacloprid, thiamethoxam, clothianidine and dinotefuran, represent approximately 25% of all insecticides in use [7,8]. This class of insecticides share a similar mode of action; they act on the central nervous system of insects as agonists for the nicotinic acetylcholine receptor (nAChRs), modulating or blocking the transmission of information causing paralysis and death of the insect [9,10]. So far, several authors have voiced alarm regarding the negative impact of neonicotinoids on non-target organisms [11]. A report published by the European Food Safety Authority (EFSA) confirmed that neonicotinoids are harmful to populations of pollinators (e.g., honeybees) [12]. As a result, the European Commission has gradually banned the use of imidacloprid, clothianidin and thiamethoxam neonicotinoids [13]. Nevertheless, these compounds are still in use in some countries [14]. In addition to the potential toxicity risk to pollinators, recent studies have shown that neonicotinoids can cause reproductive and neurological toxicity, immunotoxicity and hepatocarcinogenicity in mammals [15], and they are also considered endocrine disruptors [16].

Among neonicotinoids, main priority is given to thiamethoxam (TMX), which is commonly used to control pests such as thrips, whiteflies, aphids, leaf miners and beetle species that affect rice, maize, mango, cotton and vegetables [17]. TMX (3-[(2-chloro-1,3-thiazol-5-yl)methyl]-5-methyl-1,3,5-oxadiazinan-4-ylidene]nitramide) is a thianicotinyl subclass neonicotinoid insecticide of the second generation, introduced in 1997, and represents the major active principle of the commercial insecticide Actara 25 WG [18,19]. Like all neonicotinoids, TMX has a low capacity to sorb onto soil particles due to its low octanol water partition coefficient (log Koc = 1.75) [20], and it is very persistent in the soil (high half-life from 46 to 301 days) [21]. A study conducted in North America by Hladik et al. indicated that, after several years of cessation of treatment with the insecticides clothianidin and TMX, both were detected in soil at an average concentration of 6 ng g^−1^ [22]. Over time, soil microbial biodiversity could be compromised by TMX accumulation [23]. TMX exhibits the highest water solubility of those neonicotinoid insecticides (4100 mg L^−1^ at 25 °C) [24]. Several studies have reported the occurrence of TMX in water around the world. In the U.S., TMX was detected at concentrations of 0.05 μg L^−1^ in 67% of samples collected from the central Wisconsin groundwater [25], and in a concentration range of 0.24 and 4.15 ng L^−1^ in finished drinking water samples collected from taps at the University of Iowa, United States [26]. The three insecticides imidacloprid, clothianidin and TMX were detected at concentrations above the Canadian guidelines (230 ng L^−1^) in 75% of the samples from a stream [27]. In the Yangtze River, China, TMX was among the most frequently detected neonicotinoids, with an average concentration of 1.10 ng L^−1^ [28]. In a wastewater treatment plant in Bucharest, Romania, TMX was detected in concentrations ranging from 16.4 to 23.6 ng L^−1^ in raw sewage, while, in treated sewage, TMX was detected at average concentrations of 14.6 ng L^−1^ [29].

Growing concerns about the risks posed by TMX against non-target organisms and the quality of water resources have prompted researchers to focus on the removal of this insecticide [30]. The majority of publications reporting the degradation of TMX from water refer to Advanced Oxidation Processes (AOPs), ultraviolet radiation (UV), ozonation (O_3_) and hybrid processes UV/O_3_ [31], electrochemical oxidation [30], Fenton oxidation [32] and ultrasound oxidation [24]. Neonicotinoids are not easily biodegradable and only a few single microbial species have been reported for the biotransformation of TMX [20], such as the bacteria *Pseudomonas* sp. 1G [33], *Stenotrophomonas maltophilia* [34], *Bacillus* strains [35], *Ensifer adhaerens* TMX [36], *Bacillus* and *Pseudomonas* strains [37], *Alcaligenes faecalis*, *Escherichia coli* and *Streptococcus lactis* [38], and the fungus *Phanerochaete chrysosporium* [39]. A microbial consortium composed of *Kocuria*, *Paraburkholderia*, *Paenibacillus* and *Rhodotorula* was able to degrade TMX [40]. The nitroreduction metabolic pathway has been reported as the main process involved in the microbial transformation of TMX, contributing to the formation of four major metabolites, namely nitrosoguanidine/nitrosamine, amino-guanidine, desnitro/guanidine/imine and urea [33,36].

The finding of microorganisms able to biodegrade TMX presents an opportunity for the bioremediation of sites contaminated with this insecticide [41]; nevertheless, an accumulation of intermediate products may occur. In this study, the degradation pathway of TMX by the bacterial strain *L. portucalensis* F11 was investigated. Biodegradation intermediates were identified and the ecotoxicity of resulting degradation solutions was evaluated in order to assess the environmental risks of bacterial degradation products. *L. portucalensis* F11 revealed to be able to efficiently degrade TMX under various conditions, and of particular relevance is the fact that degradation of TMX as the only carbon and sulfur source was achieved, which has not been determined in the literature.

## 2. Results

### 2.1. Biodegradation of TMX by L. portucalensis F11

The results of the biodegradation of TMX by *L. portucalensis* F11, as presented in Figure 1, reveal that *L. portucalensis* F11 was able to degrade 41% of TMX (10.8 mg L^−1^) when supplied as the sole carbon and nitrogen source, and 35% when the same concentration was supplied as the sole carbon and sulfur source. At the same initial concentration (10.8 mg L^−1^), TMX was fully degraded by *L. portucalensis* F11 within 30 and 8 days as the single carbon source and in the presence of sodium acetate, respectively (Figure 1a). The data in Figure 2 reveal that biodegradation at increasing concentrations, 37.4, 67.6 and 128.7 mg L^−1^ of TMX as the sole carbon source, occurred at 43%, 29% and 23%, respectively (Figure 2a). Periodic feeding with sodium acetate resulted in 100%, 80% and 48% TMX degradation, respectively (Figure 2b). After 30 days of incubation, there was no TMX removal in flasks for abiotic and adsorption controls (Figure 1a).

The results in Figure 1b demonstrate that the growth of *L. portucalensis* F11 on TMX at a concentration 10.8 mg L^−1^, either as the sole carbon and nitrogen source or as the sole carbon and sulfur source, was very low (Figure 1b). Higher growth was observed when TMX was supplied as the sole carbon source in MM with nitrogen and sulfur (Figure 1b, Appendix A). Periodic feeding with 5.9 mM of sodium acetate resulted in an increase in cellular growth due to the presence of a supplementary carbon source (Appendix A). Nevertheless, it is interesting to note that TMX addition resulted in a significantly higher growth compared to that observed with only acetate feeding (Appendix A). The observed results are clear indications that the degradation of TMX contributes to the growth of *L. portucalensis* F11.

TMX degradation rate constants by *L. portucalensis* F11 were well fitted to the first-order kinetics (the value of R^2^ = 0.98 to 0.99). The degradation kinetics of TMX in the presence of sodium acetate were higher than those in its absence; this was reflected in the rate constant *k* values which increased while the corresponding half-life t_1/2_ values decreased (Table 1 and Table 2). The *k* values for TMX degradation significantly decreased and the t_1/2_ value increased when the concentrations of the insecticide increased (Table 2).

TOC and TN were measured before and after TMX degradation. Within 30 days of experimentation, total degradation of TMX as the sole carbon source (10.8 mg L^−1^) was accompanied by more than a 65% decrease in TOC content (Appendix A). This reduction could be due to the assimilation of carbon by *L. portucalensis* F11 for its growth and/or mineralization of carbonaceous matter CO_2_ during the biodegradation process. In the same conditions, the reduction in TN content reached more than 29% (Appendix A).

### 2.2. Identification of Transformation Products of TMX

The detection and identification of biodegradation products was performed by UPLC-QTOF/MS/MS. The approach to identify degradation intermediates was to screen the total ion chromatogram, acquired in full-scan mode. Selection of the chromatographic peaks originating from TMX biodegradation from those of microbial metabolism was made by comparing the mass chromatograms of the samples from the experiments inoculated with active bacterial cells and TMX to those of the control experiments. The compounds which were not detected in the controls, as well at the beginning of the experiment, were considered as potential products of TMX biodegradation.

Following this protocol, 12 intermediate metabolites were selected; their structures are depicted in Figure 3. Fragmentation patterns, detected with the MS2 equipment, allowed for the elucidation of the structural formula of each metabolite (Table 3, Appendix A). The identity of 10 of the metabolites was validated by previous reports in the literature.

Most selected intermediates present a fragment at *m/z* 132 as the parent compound, indicative of the presence of the 2-chloro-5-methyl thiazole. Considering the mass decrease of TMX-1 in relation to the parent compound, the occurrence of nitro reduction of the nitroimino group to nitrosoimino is suggested to occur. The graph of the area as a function of time shows that the concentration increases very fast, reaching a maximum at Day 1 (Figure 4).

For TMX-2, it can be stated that its mass decrease of 14 in relation to TMX-1, and the empirical formula proposed in Table 3, corresponds to a loss of one oxygen, and a concurrent gain of two hydrogens. The appearance of this metabolite is also very fast, and the profile observed is very similar to TMX-1 (Figure 4).

The *m/z* decrease between TMX-1 or TMX-2 and TMX-3 is indicative of loss of NO or NH_2_ group, respectively, followed by the replacement of the -NH by oxygen in the origin of TMX-4. The parent and fragment ions observed are in accordance with the structure of TMX guanidine and TMX urea, respectively. The formation of TMX-3 increased when the concentration of TMX-1 and TMX-2 decreased (Figure 4).

Between TMX-4 and TMX-7 an *m/z* decrease of 14 was observed, indicative of demethylation. For TMX-5 and TMX-6, it can be stated that its mass, empirical formula and fragmentation pattern corresponds to the transformation products previously reported [43]. The concentration of TMX-7 and TMX-5 decreased after Day 12 (Figure 4), while for TMX-6, the areas increased after Day 8 (Figure 4).

TMX-8 presented a decrease in molecular mass of 18 in relation to the parent compound and the fragment at *m/z* 132, indicative of the presence of the 2-chloro-5-methyl thiazole, was not present, but fragment *m/z* 115 was present. According to its fragmentation and the proposed elemental composition (Table 3), TMX-8 might result from hydroxylation at a chlorine-substituted site (Figure 4).

TMX-9 presented an elemental composition and fragmentation pattern that was compatible with the cleavage of the oxodiazine, converting TMX into clothianidin. The *m/z* difference between TMX-9 and TMX-11 and the empirical formula proposed were indicative of demethylation reaction for the formation of TMX-9. The formation pattern of both metabolites was quite similar; appearance was very fast and started decreasing after Day 12 (Figure 4).

Between TMX-9 and TMX-10, an *m/z* decrease of 46 was observed, indicative of denitrification (loss of –NO_2_ group) from clothianidin.

The *m/z* difference between TMX-10 and TMX-12 was indicative of the transformation of nitroimino moiety to the urea compound, accomplished by demethylation. The formation of TMX-12 was delayed, with the area increasing after Day 8 (Figure 4).

### 2.3. Toxicity Tests

The toxic effects produced by TMX and its metabolites were evaluated by monitoring changes in whole sample toxicity on *Lactuca sativa*, *Aliivibrio fischeri* and using a Toxi-Chromo test. The controls with medium supplemented with sodium acetate not inoculated and inoculated with *L. portucalensis* F11 degrading strain had no toxic effect on the tested organisms.

*Lactuca sativa* seed germination, root and shoot elongation were not affected when exposed to TMX (10.8 and 37.4 mg L^−1^) or to its degradation products. On the contrary, TMX supplied in concentrations of 10.8 mg L^−1^ and 37.4 mg L^−1^ decreased the luminescence emitted by *Aliivibrio fischeri* by 19% and 28%, respectively. While the samples from the end of the biodegradation experiments had no inhibitory effects on this organism. Similar observations were noted using the Toxi-Chromo test. The neonicotinoid TMX inhibited the synthesis of β-galactosidase by *E. coli* by 7.4% for the concentration 10.8 mg L^−1^, and 12.0% for the concentration 37.4 mg L^−1^. Synthesis of β-galactosidase was found to be unaffected by the exposure to final samples, as revealed by conversion of the chromogenic substrate at levels similar to the negative control (Appendix A). These results confirmed a removal in whole sample toxicity at the end of the biodegradation experiments.

## 3. Discussion

### 3.1. Biodegradation of TMX by L. portucalensis F11

TMX is considered a potential contamination source of surface and ground waters [30] and, due to this, was included on the European First and Second Watch Lists for emerging water pollutants [45,46]. *L. portucalensis* F11 was able to achieve complete degradation of TMX (10.8 mg L^−1^) supplied as the sole carbon source over 30 days. The supplementation with acetate allowed complete degradation of the same concentration in 8 days.

The degradation of TMX as the sole carbon source by *L. portucalensis* F11 was in the same order of magnitude as in previous reports. Rana et al. reported that *Bacillus aeromonas* strain IMBL 4.1 and *Pseudomonas putida* strain IMBL 5.2 were able to degrade 45.28% and 38.23%, respectively, of the supplied amount of TMX (50 mg L^−1^) within 15 days of incubation at 37 °C as the sole carbon source [37]. Furthermore, the *endophytic Enterobacter* cloacae TMX-6 isolated from *Ophiopogon japonicas* was able to degrade 20% from the initial concentration of TMX (10 mg L^−1^) as the sole carbon source after 15 days of incubation at 30 °C [47]. Boufercha et al. found that five *Streptomyces*. sp strains isolated from a wastewater treatment plant were able to degrade TMX (35 mg L^−1^) at a low rate ranging from 2% to 19% after 30 days of incubation as the sole carbon source [48].

It is known that co-metabolism plays an important role in the microbial transformation of pesticides [49], and the process can be very fast in the presence of supplementary carbon sources [50]. Accordingly, the results of the present study revealed faster and efficient degradation of TMX in the presence of a supplementary carbon source, which allowed higher cell density. Excluding the beneficial effect of acetate addition, *L. portucalensis* F11 demonstrated the use of TMX for growth. TMX degradation during co-metabolism was also described in a study conducted by Pandey et al. [33]. They reported that the addition of 10 mM of glucose allowed *Pseudomonas* sp. 1G to degrade 70% of both TMX and imidacloprid insecticides (50 mg L^−1^) after 14 days of incubation at 28 °C and under micro-aerophilic conditions [33]. In a study conducted by Myresiotis et al., bacteria from *Bacillus* genus presented an 11% to 22% reduction of 0.25–2.5 mg L^−1^ TMX over 3 days in TSB medium [35]. When cultured in half-strength nutrient broth supplemented with 70 mg L^−1^ TMX, reductions in TMX concentration were observed for *P. fluorescens* (67%), *P. putida* (65%), *P. aeruginosa* (52%) and *A. faecalis* (39%) over 24 days [38]. A recent study conducted by Boufercha et al. showed that TMX degradation by *Streptomyes* sp. OV and *Streptomyces* sp. OB increased when glucose (10 mM) or sodium acetate (5.9 mM) were supplemented, respectively [48]. In the present study, degradation rates decreased with increasing TMX concentration. Chen et al. observed similar results, as they found that white-rot fungus *Phanerochaete chrysosporium* was able to eliminate 98%, 74%, and 27% of the initial concentration 10, 20, and 50 mg L^−1^ of TMX after 25 days of incubation 37 °C [39].

*L. portucalensis* F11 showed an ability to degrade TMX as a single source of carbon and nitrogen, and as a sole carbon and sulfur source. There have few studies evaluating the degradation of TMX as a sole carbon and nitrogen source. Zhou et al. found that the rhizobacterium *Ensifer adhaerens* strain TMX-23, isolated from rhizosphere, was able to eliminate 16.3% from 10 mg L^−1^ of TMX after 5 days of incubation [36]. An imidacloprid-degrading consortium removed 33.4% thiamethoxam (50 mg L^−1^) in Bushnell-Haas lacking N source in 31 days [40]. In contrast, Pandey et al. found that *Pseudomonas* G1 was unable to use TMX as a sole carbon and nitrogen source [33]. The present study is the first to use TMX as a single carbon and sulfur source.

It is essential to highlight that the degree of mineralization, i.e., the conversion of a chemical into inorganic substances or compounds usable by microorganisms to produce cellular biomass, is a crucial parameter in the biodegradation of organic pollutants [51]. Complete removal of TMX as the sole carbon source was accompanied by more than a 65% of TOC decrease during the time course of the experiment. The residual TOC can be attributed to some metabolites still present at the end of the experimental period (Figure 4), and possible soluble polymers from the biomass. Cui et al. found that, during the biodegradation of quinoline by *Comamonas* sp., 70% of the TOC was removed [52]. Moreover, the achieved TOC removal was much higher than that reported for the photocatalytic degradation of TMX [53]. In relation to the metabolites detected during the degradation of TMX as the sole carbon source, most of them revealed a tendency to decrease toward the end of the experimental period, even those for which their formation was at latter stages, suggesting further degradation.

The microbial biodegradation of pollutants is influenced by a variety of environmental factors [54]. *L. portucalensis* F11 is Gram-negative, non-spore-forming and non-motile aerobic bacterium isolated from contaminated sediment in northern Portugal [55,56]. This bacterial strain can grow in a wide range of temperatures (16 °C–37 °C) and pH (4.0–8.0) [55], which confers a great potential for adaptation to various environmental conditions. In the present study, *L. portucalensis* F11 was evaluated for the first time for its ability to degrade a pesticide. Nevertheless, this bacterium has shown impressive degradation abilities towards pharmaceuticals such as fluoroquinolones (ofloxacin, norfloxacin and ciprofloxacin) [57], fluoxetine [58], diclofenac [59], and carbamazepine [60] and aromatics compounds, such as fluorobenzene [55,56], difluorobenzenes [61], and fluoroaniline [62], and. *L. portucalensis* F11 is a rhizobial bacterium that can potentially be used as an inoculant in soil to improve the absorption of minerals (N, P, K, Fe, Zn, and Mg) required for plant growth and to bioremediate soils contaminated with TMX.

### 3.2. TMX Degradation Pathway

Based on the intermediates identified, three routes of TMX biodegradation by *L. portucalensis* F11 are proposed. A major metabolic pathway of TMX biodegradation involves the transformation of its N-nitroimino group (=N-NO_2_) to N-nitrosimine/nitrosoguanidine (=N-NO), amino-guanidine (=N-NH_2_), desnitro/guanidine/imine (=NH), and urea (=O) metabolites. This route has been reported in previous studies [33,36,42]. The urea metabolite was further metabolized by *L. portucalensis* F11 through demethylation. A second pathway was detected with the cleavage of oxadiazine, resulting in the conversion of TMX into clothianidin, as reported by Zhou et al., for soil microorganisms [42]. The pathway of oxadiazine ring cleavage resulting in the generation of clothianidin is not exclusive to microorganisms; it has also been previously reported in mice, insects, spinach, and tomatoes [63]. Further clothianidin biodegradation by *L. portucalensis* F11 proceeds through denitrification, as previously observed in *Pseudomonas stutzeri* smk [44]. In parallel, we also observed the demethylation of clothianidin, followed by the transformation of the nitroimino moiety to urea, similar to the mechanism observed in the degradation of clothianidin by a bacterial consortium [64], but, in the present study, the formation of the metabolite was preceded by a demethylation reaction. Two more metabolites, which could be attributed to a cleavage of the oxodiazine ring, were detected, but the cleavage was in a different position to the one that gives origin to clothianidin. These metabolites were previously identified in wastewater [43]. Finally, the third pathway observed was the dechlorination of TMX, which has recently been reported in the metabolic pathway of the white-rot fungus *Phanerochaete chrysosporium* [39]. A hydrolytic dehalogenation mechanism replacing chlorine with a hydroxyl group was previously identified in *L. portucalensis* F11 during biodegradation of the pharmaceuticals ciprofloxacin [57] and diclofenac [59].

### 3.3. Toxicity Tests

The risk of producing metabolites with higher toxicity than the parent compound was evaluated by monitoring changes in whole sample toxicity. In the present study, three bioassays were used to evaluate the evolution of the toxicity of TMX degradation samples. *Lactuca sativa* was used as a biomarker of phytotoxicity. The seed germination, root and shoot elongation were not sensitive to the presence of TMX, in accordance with the previously report by Chen et al., with rape and cabbage [39]. Likewise, Macedo and Castro reported that TMX stimulated the growth of rice [65]. The toxicity of the TMX was also evaluated towards the marine bacterium *Aliivibrio fischeri*. It was found that TMX decreased the bioluminescence of the bacterium, reflecting the toxic effect of this compound. Similar results were observed by Šojić et al., when TMX was applied at an initial concentration of 0.05 mmol L^−1^ [31]. Moreover, the synthesis of β-galactosidase was also slightly inhibited by TMX. There has been no previous study on the toxicity of TMX using the Toxi-Chromo test, but Chen et al. indicated that TMX at a concentration 2.5 mg L^−1^ showed little influence on the growth of *E. coli* [39]. The exposure to final samples of TMX biodegradation by *L. portucalensis* F11 did not affect the seed germination, the bioluminescence of *Vibrio fischerim*, and the production of β-galactosidase by *E. coli*. These results revealed that biodegradation of TMX by *L. portucalensis* F11 was accompanied by a removal of toxicity.

## 4. Materials and Methods

### 4.1. Chemicals and Materials

During this study, ultrapure water with a resistivity of 18.2 M Ω cm was produced by a Milli-Q Gradient A-10 system (Millipore). The commercial TMX formulation (25% *w/w*) was provided by Syngenta, while the active principle TMX (≥98%) was purchased from Sigma-Aldrich (Steinheim, Germany). The chemical ingredients used to prepare the minimal salt medium MSM1 [66], MSM2 [67], and MSM3 [68] were of analytical grade (Sigma-Aldrich Chemie, Steinheim, Germany; Merck, Darmstadt, Germany). Sodium acetate and acetonitrile HPLC grade were provided by Merck (Darmstadt, Germany). Trifluoroacetic acid 99% was purchased from Sigma-Aldrich (Steinheim, Germany). Microfiber glass filters of 0.45 µm (Whatman™, Maidstone, UK) were used to filter HPLC grade solvents. All other chemicals and solvents used were of the highest purity and of analytical grade.

### 4.2. Culture Conditions

The bacterial strain *L. portucalensis* F11 (GenBank/EMBL/DDBJ accession number AY362040; DSMZ accession number DSM 17916) was isolated from a sediment collected from an industrially polluted zone in northern Portugal, as previously reported [56]. This strain was used due to its proven degradation abilities [58,59,60,61,69,70]. The bacterial inoculum was prepared by growing the strain *L. portucalensis* F11 on nutrient agar medium. After incubation for 2 days at 30 °C, cells were suspended in sterilized MSM and adjusted to a final OD of 0.1 (λ = 600) for biodegradation experiments.

### 4.3. Biodegradation Experiments

Biodegradation of TMX by strain *L. portucalensis* F11 was carried out in 500 mL flasks containing 170 mL of MSM1 or MSM2, both supplemented with the insecticide at a final concentration of 10.8 mg L^−1^ as the sole carbon nitrogen source and as the sole carbon sulfur source, respectively. MSM3 medium supplemented with 10.8 mg L^−1^ of the target pollutant was used to assess the removal of this insecticide by *L. portucalensis* F11 as the sole carbon source. This concentration, although higher than that typically present in the environment, allowed us to follow TMX degradation and detect metabolites, and it was in the same order of magnitude as that used in similar studies [47]. Sodium acetate was employed as a supplementary carbon source, supplied at 5.9 mM in order to simulate the carbon content of an influent of a WWTP [71]. Periodic feeding with a constant concentration of 5.9 mM on days 5, 10, 15, 20, and 25 was performed. The effect of the initial concentration of TMX on the degradation potential of *L. portucalensis* F11 was examined in the range from 37.3 to 128.7 mg L^−1^, in the same order of magnitude as used in similar studies [37,48], and as the sole carbon source and in co-metabolism with sodium acetate.

According to the conditions of each experiment, biotic and abiotic controls were established. In addition, MSM inoculated with *L. portucalensis* F11 inactivated by autoclaving served as the adsorption control. All experiments were performed in triplicate. The flasks were wrapped in aluminum paper and incubated in the dark to avoid photolytic degradation. The cultures were incubated under agitation (130 rpm) at 30 °C for 30 days. Samples of the different experiments were periodically collected at defined intervals and plated on nutrient agar plates to check the purity. To assess cell growth, the optical density at 600 nm was measured with a spectrophotometer, type UNICAM- Hλeios.

The first-order kinetic model was used to calculate the degradation rate of TMX according to Equation (1).
C = C_0_ × e^−*kt*^(1)

Within Equation (1), C_0_ is the concentration of TMX at *t*_0_, C is the concentration of the insecticide at time *t*, and *k* is the degradation rate constant (day^−1^).

The half-life biodegradation (*t*_1/2_) was calculated using the following Equation (2).
*t*_1/2_ = ln2/*k*(2)

### 4.4. Analytical Method

#### 4.4.1. TOC-TN Analysis

Total Organic Carbon (TOC) and Total Nitrogen (TN) were measured using a TOC-VCSH/CSN instrument (Shimadzu, Kyoto, Japan). The supernatant of selected samples was previously acidified with sulfuric acid in order to attain a pH between 2–3.

#### 4.4.2. TMX Analysis

Residual TMX was quantified by HPLC in the supernatant recovered from the biodegradation experiments after centrifugation at 8000× *g* for 10 min at 4 °C. Separation was carried out using a reversed phase HPLC Cartridge LiChrospher 100 RP-18 column (Merck), coupled to a Beckman Coulter System Gold 126 and a UV detector. The chromatographic conditions were as previously reported [48].

#### 4.4.3. Identification of Degradation Products

The biotransformation products were detected and identified on an UltiMate 3000 Dionex UHPLC (Thermo Scientific, Waltham, MA, USA), coupled to an Ultra-High Resolution Qq-Time-Of-Flight (UHR-QqTOF) mass spectrometer with 50,000 Full-Sensitivity Resolution (FSR) (Impact II, Bruker Daltonics, Bremen, Germany).

Separation of metabolites was performed using an Acclaim RSLC 120 C18 column (100 mm × 2.1 mm, 2.2 µm) (Dionex). Mobile phases were 0.1% aqueous formic acid (solvent A) and acetonitrile with 0.1% formic acid (solvent B). The gradient started with 10% B, increased to 21% in 10 min, then to 27% in up to 14 min followed by 58% until 18.3 min; in the end, it increased to 100% until 20 min, after which it stayed constant during 4 min; after that, it returned to 10% B and was maintained at 5% B for an additional 2 min at a flow rate of 0.25 mL min^−1^. The injection volume was 5 µL. The parameters for MS analysis were set using positive ionization mode with spectra acquired over a range from *m/z* 50 to 1000. The parameters were as follows: capillary voltage, 4.5 kV; drying gas temperature, 200 °C; drying gas flow, 8.0 L min^−1^; nebulizing gas pressure, 2 bar; collision RF, 300 Vpp; transfer time, 120 µs; and prepulse storage, 4 µs. Post-acquisition internal mass calibration used sodium formate clusters with the sodium formate delivered by a syringe pump at the start of each chromatographic analysis.

High-resolution mass spectrometry was used to identify possible intermediates. The elemental composition of the compounds was confirmed according to accurate masse and isotope rate calculations designated as mSigma (Bruker Daltonics). The accurate mass measured was within 5 mDa of the assigned elemental composition and mSigma values of <20 provided confirmation. The structure of each metabolite was elucidated using MS2 product ion scan.

### 4.5. Toxicity Tests

For toxicity assays, samples were withdrawn from the experiments in which total TMX removal was obtained at 0 and 30 days without dilutions. The biomass was removed by centrifugation at 14,000× *g* rpm for 10 min at 4 °C and kept at −20 °C until the moment of analysis. The pH was not adjusted. Samples were analyzed without dilution in order to assess the evolution of whole sample toxicity and to compare the final toxicity with the control experiments. All the assays were carried out in quadruplicate.

Seeds of *Lactuca sativa* (lettuce) were used as a model crop in the phytotoxicity test. The inhibition of seed germination and the inhibition of root and shoot length were determined by comparison to the controls obtained by exposing seeds to distilled water according to OECD Guideline 208 [72]. The seeds were surface sterilized with bleach solution (5% commercial bleach) for 15 min and washed three times with sterile water. A total number of 10 seeds were placed in Petri dishes containing Whatman No. 2 filter paper and exposed to 3 mL of samples; the dishes were sealed with Parafilm in an incubator at 25 °C in the dark. The experiments were performed for 7 days. After that period, the seedlings were visually checked in order to count the germinated seeds and measure root and shoot growth. The inhibition normalized on negative control data was expressed as a percentage. Seedling emergence of 80% was observed in the control performed with distilled water, proving seed viability.

The *Aliivibrio fischeri* luminescence test was performed according to UNI EN ISO 11348-2-2007 [73] and following the Luminescent Bacteria Test LCK 480 manufacturer’s instructions (Hach Lang GmbH, Dusseldorf, Germany). This method is based on the percentage of inhibition of light emitted by the bioluminescent bacterium *Aliivibrio fischeri* upon contact during 15 min with the sample. Toxi-Chromo test is a toxicant-sensitive bacterial kit. The experimental procedure of this test was based on the protocol described by the manufacturer’s instructions (EBPI, Mississauga, ON, Canada). In mutant *E. coli* (K12 OR85), the potential for the inhibition of the de novo synthesis of an inducible β-galactosidase was determined. The activity of the enzyme was detected by the hydrolysis of a chromogenic substrate.

## 5. Conclusions

The bacterial strain *L. portucalensis* F11 showed efficient degradation of TMX under various conditions, accompanied by toxicity removal, and exhibited desirable attributes that could be exploited in bioremediation applications. Nitro reduction, oxadiazine ring cleavage, and dechlorination were the main degradation pathways proposed based on the identified metabolites. Results from the present study revealed that *L. portucalensis* F11 can potentially be used as part of a TMX bioremediation technology for a simple, ecologically-friendly and sustainable way to remove TMX from the environment.

## Figures and Tables

**Figure 1 ijms-23-14326-f001:**
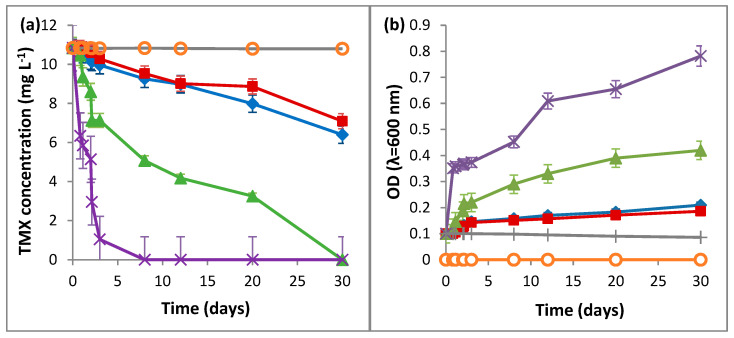
Use of TMX by *L. portucalensis* F11 at a concentration 10.8 mg L^−1^ during 30 days of incubation, as (♦) the sole carbon and nitrogen source; (■) the sole carbon and sulfur source; (▲) the sole carbon source; (**×**) periodic feeding with 5.9 mM of sodium acetate; abiotic (○) and adsorption (+) controls. (**a**) TMX degradation and (**b**) cellular growth. Error bars present means of three replicates ± standard deviation.

**Figure 2 ijms-23-14326-f002:**
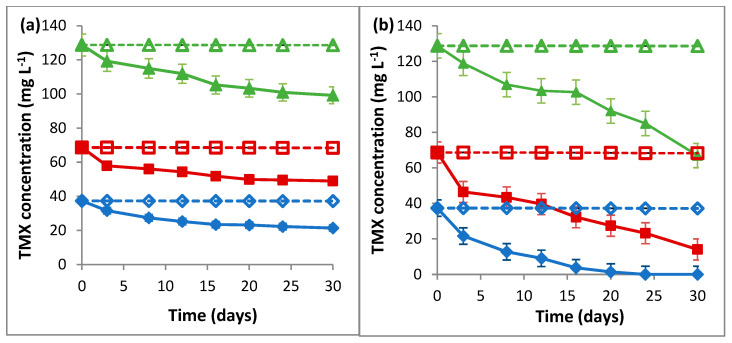
Biodegradation of TMX by *L. portucalensis* F11 at concentrations: 37.4 mg L^−1^ (♦), 68.6 mg L^−1^ (■) and 128.7 mg L^−1^ (▲), and respective abiotic controls (◊, ⸋, Δ), during 30 days of incubation, (**a**) as the sole carbon source and (**b**) with periodic feeding with sodium acetate (5.9 mM). Error bars present means of three replicates ± standard deviation.

**Figure 3 ijms-23-14326-f003:**
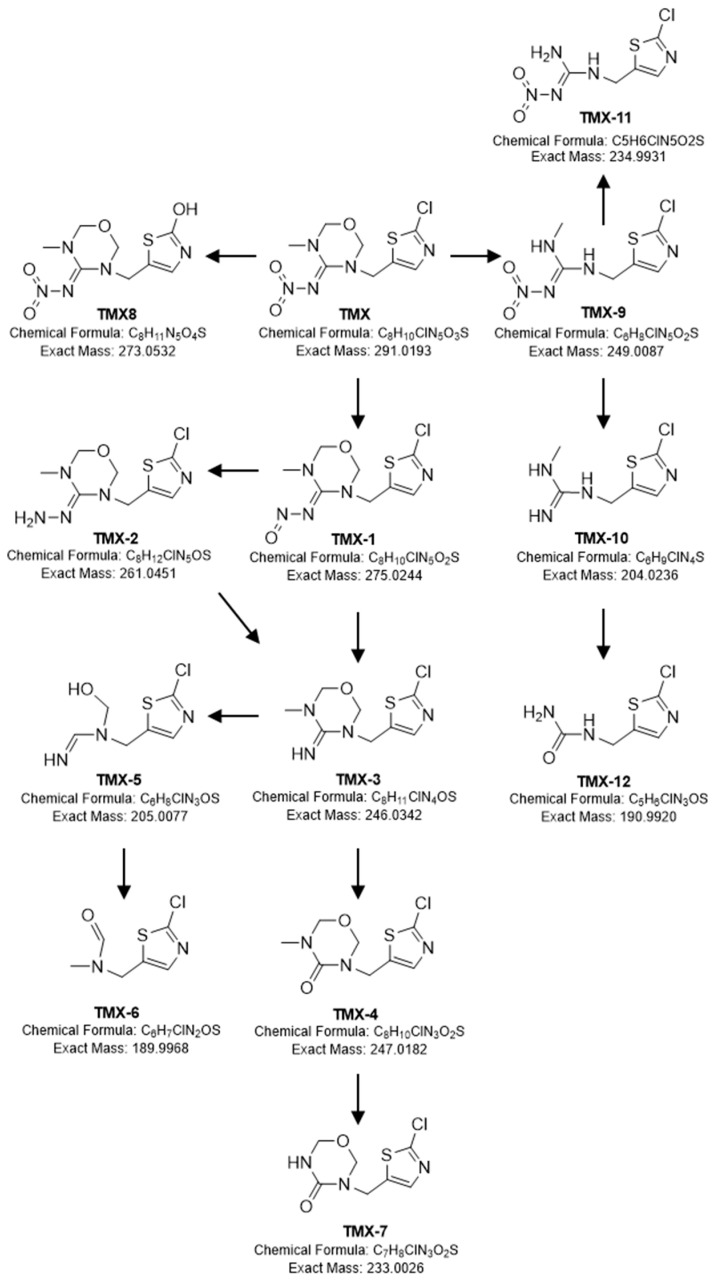
Proposed TMX biodegradation pathway by *L. portucalensis* strain F11.

**Figure 4 ijms-23-14326-f004:**
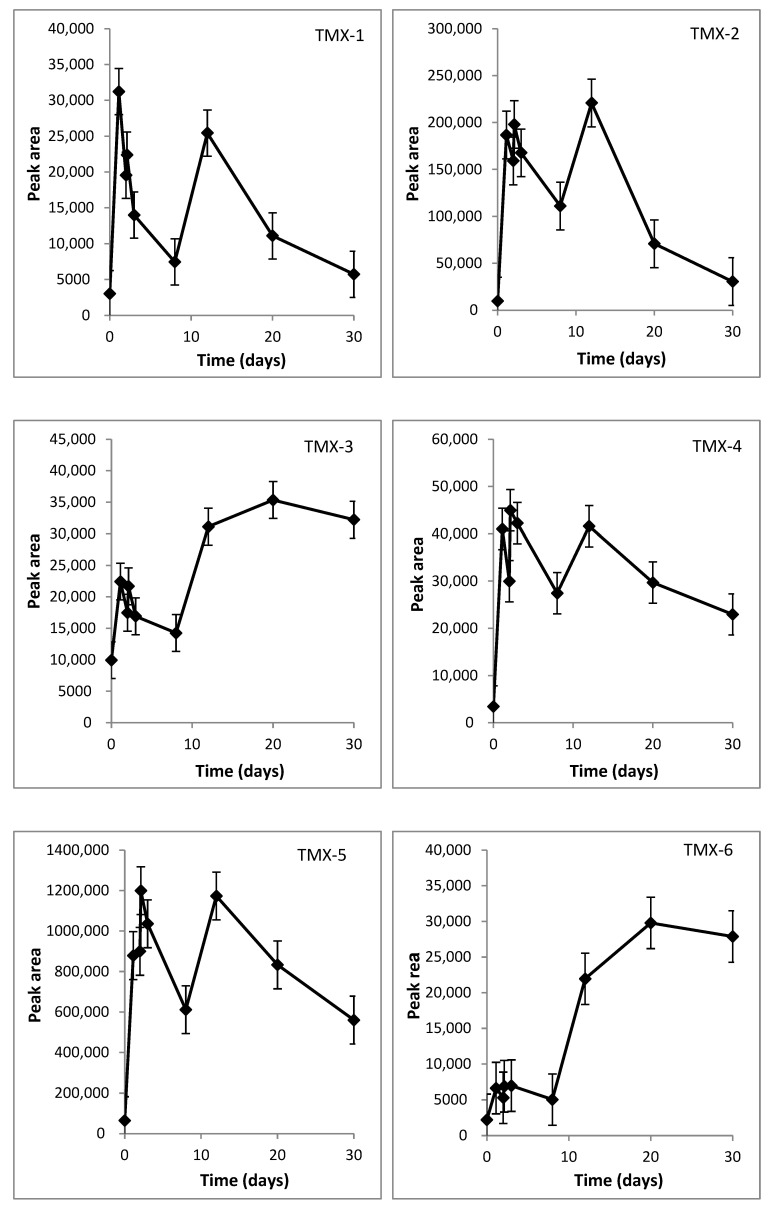
Peak area formation of the intermediate metabolites TMX-1–TMX-12 during TMX degradation as the sole carbon source by *L. portucalensis* F11 in function of time.

**Table 1 ijms-23-14326-t001:** First-order rate constant (*k*) and half-life (t_1/2_) for degradation of TMX (10.8 mg L^−1^) by *L. portucalensis* F11.

Biodegradation of TMX as:	*k* (d^−1^)	t_1/2_ (d)	R^2^
Carbon and nitrogen source	0.017 ± 0.002	40.77 ± 0.129	0.9904
Carbon and sulfur source	0.010 ± 0.002	69.31 ± 0.091	0.9836
Carbon source	0.077± 0.002	8.961 ± 0.015	0.9966
Supplemented with acetate	0.791 ± 0.002	0.872 ± 0.001	0.9974

**Table 2 ijms-23-14326-t002:** First-order rate constant (*k*) and half-life (t_1/2_) for degradation of TMX at different concentrations by *L. portucalensis* F11.

TMX Concentration (mg L^−1^)	TMX as Sole Carbon Source			TMX with Periodic Feeding with Acetate		
	*k* (d^−1^)	t_1/2_ (d)	R^2^	*k* (d^−1^)	t_1/2_ (d)	R^2^
37.3	0.022 ± 0.0015	31.363 ± 0.005	0.9925	0.156 ± 0.0015	4.423 ± 0.0015	0.9895
68.6	0.016 ± 0.0009	43.125 ± 0.007	0.9933	0.053 ± 0.0004	13.018 ± 0.0054	0.9941
128.7	0.010 ± 0.0020	69.000 ± 0.150	0.9915	0.023 ± 0.0014	31.000 ± 0.0550	0.9994

**Table 3 ijms-23-14326-t003:** TMX degradation metabolites detected by UPLC-QTOF/MS/MS.

Metabolite	tr (min)	Measured *m/z*	Products MS2	Empirical Formula	References
TMX-1	8.0	276.0144	131.9574, 181.0392, 100.0698, 69.0398	C_8_H_10_ClN_5_O_2_S	[33,36,42]
TMX-2	8.1	262.0161	131.9667, 70.9950	C_8_H_12_ClN_5_OS	[43]
TMX-3	4.4	247.0412	131.9665, 160.9940, 44.0488	C_8_H_11_ClN_4_OS	[33,42,43]
TMX-4	11.2	248.0083	174.9607, 132.9650	C_8_H_10_ClN_3_O_2_S	[33,36,42,43]
TMX-5	6.9	206.0156	174.9729, 113.0172, 86.0058, 58.9941	C_6_H_8_N_3_OSCl	[43]
TMX-6	2.7	191.0032	131.9593, 162.8899, 70.9900	C_6_H_7_ClN_2_OS	[43]
TMX-7	8.3	234.0102	174.9737, 131.9673	C_7_H_8_ClN_3_O_2_S	[43]
TMX-8	8.3	274.0414	137.0723, 115.0664, 84.0490, 69.0398	C_8_H_11_N_5_O_4_S	[39]
TMX-9	10.4	250.0171	131.9674, 169.0543, 70.9951	C_6_H_8_ClN_5_O_2_S	[43]
TMX-10	21.8	205.0172	131.9587, 166.9478, 113.0102	C_6_H_9_ClN_4_S	[43,44]
TMX-11	8.5	235.9951	131.9577, 174.9603, 125.0087	C_5_H_6_ClN_5_O_2_S	This study
TMX-12	5.3	191.9868	131.9591, 174.9595, 70.9912	C_5_H_6_ClN_3_OS	This study

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
