# Peer review of "Biodegradation and Metabolic Pathway of the Neonicotinoid Insecticide Thiamethoxam by Labrys portucalensis F11"

_ijms, 2022, doi:10.3390/ijms232214326_

Round 1

Reviewer 1 Report

The manuscript entitled Biodegradation and metabolic pathway of the neonicotinoid insecticide thiamethoxam by Labrys portucalensis F11 submitted by Bouferchaet al. investigates biotransformation of neonicotinoid insecticide thiamethoxam by Gram negative bacterium Labrys portucalensis F11. The results are within the scope of IJMS. Biodegradation of TMX has been extensively studied by other authors so the paper (in my opinion) does not present significant novelty. The quality of figures must be improved. My review comments that should be addressed before publication: 

Line 94-99: The novelty of the study should be highlighted. The authors should provide more details about ecotoxicology experiments.

Line 102: Biodegradation is not shown in Figure 1. The results of biodegradation experiments are presented in Figure 1. The authors should correct this in the entire manuscript.

Line 111: The authors should provide data regarding control samples. 

Line 126: What was the reason of choosing such concentrations of TMX? 

Line 199: The quality of Fig 3 must be improved.

Fig 4 contains too many figures. These data should be compressed to present the most important results of the experiments. The Y axes should have identical scale. At the present form, the result are very confusing.

Line 443: Conclusion section must be rewritten to highlight research hypothesis or research question and major findings. Also, the reader must know what contribution the study has made to the existing literature.

Reviewer 2 Report

In the paper, the author pointed Biodegradation and metabolic pathway of the neonicotinoid insecticide thiamethoxam by Labrys portucalensis F11. I strongly recommend this article is worthy to publish in the journal after correction of some points described below:

1.       The aim of the work should be redrafted. The aim of the study cannot be to determine the ability of Labrys portucalensis F11 to biodegrade TMX, because it is obvious.

2.       Fig.2. data on the experimental system without bacteria (control) are missing.

3.       The quality of signatures in Figure 3 should be improved.

4.        Line 228. Sentence L. portucalensis F11 was able to degrade TMX ”. It is obvious and dos not introduces nothing new to work.

5.       I cannot refer to supplementary materials, I do not have access to them.

Reviewer 3 Report

Thiamethoxam (TMX) is a widespread use insecticide. It is very persistent in the soil. Growing concerns about the risks posed by TMX against non-target organisms and the quality of water resources have prompted researchers to focus on the removal of it. In the manuscription, the author isolates one strain, Labrys portucalensis F11. F11 can degrade TMX (10.8 mg L-1) more than 90% was removed in 3 days. Overall, it is a well-written paper, but I have two small questions below.

1. The language needs to be checked. 

2. Please provides the figure of UPLC-QTOF/MS/MS in the text.

Round 2

Reviewer 2 Report

I accept the revised manuscript